# Biological Plausibility of Using Plasma Amino Acid Profile Determination as a Potential Biomarker for Pediatric Patients with Mild Traumatic Brain Injuries

**DOI:** 10.3390/neurolint17090145

**Published:** 2025-09-09

**Authors:** Adán Pérez-Arredondo, Eduardo Cázares-Ramírez, Luis Tristán-López, Carlos Jiménez-Gutiérrez, Diana L. Pérez-Lozano, Ivette A. Martínez-Hernández, Valentina Vega-Rangel, Hugo F. Narváez-González, Camilo Rios, Marina Martínez-Vargas, Luz Navarro, Liliana Carmona-Aparicio

**Affiliations:** 1Departamento de Fisiología, Facultad de Medicina, Universidad Nacional Autónoma de México (UNAM), Mexico City 04510, Mexico; aperez@facmed.unam.mx (A.P.-A.); marina.martinez@facmed.unam.mx (M.M.-V.); 2Programa de Maestría y Doctorado en Ciencias Médicas, Odontológicas y de la Salud, Investigación Clínica Experimental, Universidad Nacional Autónoma de México (UNAM), Mexico City 04510, Mexico; amcrr1962@gmail.com; 3Departamento de Urgencias Médicas Pediátricas, Instituto Nacional de Pediatría, Mexico City 04530, Mexico; eduardocazaresr@gmail.com (E.C.-R.); valentina.vega@facmed.unam.mx (V.V.-R.); 4Laboratorio de Neuroquímica, Instituto Nacional de Neurología y Neurocirugía Manuel Velasco Suárez, Mexico City 14269, Mexico; carbolit2001@gmail.com; 5Subdirección de Investigación Biomédica, Hospital General “Dr. Manuel Gea González”, Mexico City 14080, Mexico; 6Coordinación de Medicina General y Comunitaria, Universidad de la Salud, Campus Santa Fe, Mexico City 01210, Mexico; dianalozano3777@gmail.com; 7Escuela de Ciencias de la Salud, Universidad del Valle de México, Campus Coyoacán, Mexico City 04850, Mexico; 8Hospital “Primero de Octubre”, Instituto de Seguridad y Servicios Sociales de los Trabajadores del Estado (ISSSTE), Mexico City 73000, Mexico; ivette_17217@hotmail.com; 9Centro Médico Nacional “20 de Noviembre”, Instituto de Seguridad y Servicios Sociales de los Trabajadores del Estado (ISSSTE), Mexico City 03100, Mexico; drnarvaezg@hotmail.com; 10Instituto Nacional de Rehabilitación “Luis Guillermo Ibarra Ibarra”, Mexico City 14389, Mexico; camrios@yahoo.com.mx; 11Laboratorio de Farmacología, Instituto Nacional de Pediatría, Mexico City 04530, Mexico

**Keywords:** mild traumatic brain injury, pediatric, amino acids, glutamate, aspartate, glutamine, glycine

## Abstract

Background: Amino acid biomarkers have a crucial influence on our understanding of brain injury mechanisms, and their plasma concentrations may indicate neurological damage and recovery patterns. Pediatric mild traumatic brain injury (mTBI) assessment particularly benefits from such molecular indicators, as clinical presentations can be subtle and variable. However, current diagnostic and prognostic tools lack reliable biochemical markers that can track the temporal evolution of injuries and recovery. Methods: We conducted a prospective longitudinal cohort study involving 36 pediatric mTBI patients and 44 controls to characterize the temporal evolution of key amino acids and their derived indices. Blood samples were collected at 3, 6, 12, and 24 h and at 7, 14, and 28 days post-injury, with amino acids quantified using high-performance liquid chromatography. Results: Our analysis revealed significant temporal changes in glutamate, glutamine, and glycine concentrations, with glutamate peaking at day 7 before declining, while glutamine showed steady increases throughout. The GLN/GLU ratio demonstrated an early excitatory imbalance followed by astrocytic compensation, and the GLX ratio indicated progressive recovery. Conclusions: These patterns represent continuous neurochemical processes involving excitotoxicity and glial regulation, suggesting potential utility as biomarkers for mTBI diagnosis and monitoring. While further validation using larger cohorts is needed, these findings provide compelling evidence of the efficacy of using amino acid profiles to track pediatric mTBI progression and recovery.

## 1. Introduction

A traumatic brain injury (TBI) is a functional disturbance of the central nervous system caused by a sudden transfer of mechanical energy, which can result in temporary or permanent neurological impairment [1,2,3]. It is classified according to the Glasgow Coma Scale (GCS) as mild (13–15), moderate (9–12), or severe (3–8). Despite its limitations, the GCS remains widely used [4,5]. The American Congress of Rehabilitation Medicine and the World Health Organization define a mild TBI (mTBI) as a loss of consciousness for less than 30 min, post-traumatic amnesia for less than 24 h, acute changes in mental status, or transient neurological deficits [6].

TBIs are a significant global health concern because of their high morbidity and mortality rates, particularly among young individuals [7,8,9]. Every year, it is estimated that 69 million people sustain a TBI, with 55.9 million of these cases being categorized as mTBIs. When non-hospitalized cases are considered, the incidence rates surge to between 700 and 800 per 100,000 members of a population [6]. Despite their high prevalence, mTBIs are frequently underestimated, especially among children and adolescents. These individuals are more neurologically vulnerable, require longer recovery periods, and have an increased risk of experiencing cumulative effects from repeated injuries [10,11].

Globally, the leading causes of TBIs are traffic accidents, falls, injuries due to contact sports, and interpersonal violence. The most affected populations are young males (15–35 years) followed by older adults and children aged 0 to 4 [6,11,12]. Pediatric patients with a mild TBI often exhibit enduring symptoms such as headaches, cognitive defects, emotional disturbances, and sleep disorders [11].

TBIs are characterized by a primary, irreversible injury that occurs at the moment of impact and a potentially reversible secondary injury involving neuroinflammation, apoptosis, and synaptic dysfunction [1,13,14,15,16,17]. Early diagnosis, especially in the case of an mTBI, remains a challenge because of the lack of sensitive clinical tools that can predict functional outcomes [18].

Recent studies have shown that, after an mTBI, there is an acute increase in the permeability of the blood–brain barrier (BBB), which allows the release of neuronal and glial biomarkers into peripheral fluids such as blood, saliva, or urine. This phenomenon provides significant clinical opportunities for the early detection and longitudinal monitoring of TBIs, depending on the timing, severity, and progression of the pathological process [19]. However, these studies are conducted on the adult population, leaving a gap in information on what happens in the pediatric population.

Conventional clinical tools, such as neurological assessments, computed tomography (CT), and magnetic resonance imaging (MRI), have limitations in terms of detecting alterations in cases of mTBIs, particularly in pediatric populations [20,21,22]. Despite advances in biomarkers, such as GFAP and UCH-L1, there are still no validated tools available for routine clinical application to children [18,19,22,23,24,25,26]. Following trauma, levels of central nervous system (CNS)-derived biomarkers initially increase in cerebrospinal fluid and may become detectable in blood and other body fluids over time [27]. This phenomenon provides early diagnostic and prognostic opportunities, which can lead to clinical and therapeutic management in the pediatric population.

Although proteins such as GFAP, UCH-L1, NSE, and especially SB100 have been analyzed in regard to pediatric TBIs, they have been mainly been proposed to be biomarkers of damage in adults with moderate or severe injuries [22,24,25,26,28,29,30,31,32,33,34,35,36,37,38,39,40,41,42,43,44,45,46,47,48,49]. Their clinical application to pediatric patients is limited by methodological challenges and a lack of evidence [50]. In this context, the study of brain amino acids is a promising approach. Glutamate, the principal excitatory neurotransmitter involved in post-traumatic excitotoxicity, has been linked with worse clinical outcomes when accumulated [51,52,53,54,55,56,57]. In contrast, levels of inhibitory amino acids like GABA might increase as a compensatory response [58,59,60,61]. Investigations involving patients with severe TBIs have reported elevated plasma levels of glutamate, GABA, aspartate, glutamine, and cysteine, indicating that the amino acid profile may mirror specific neurochemical responses to trauma [62]. These findings underline amino acids’ potential as non-invasive biomarkers for assessing the progression of brain damage, especially in the pediatric population, where longitudinal monitoring is vital for immediate intervention and prognostic evaluation and where it is a priority to determine the damage sustained after a TBI with its different degrees of severity.

The clinical utility of amino acids as biomarkers in regard to pediatric mTBIs is not yet clearly established. This study evaluates the plasma levels of glutamate, aspartate, glutamine, and glycine; the ratio of glutamine to glutamate; and GLX in children with mild TBIs over a period of 30 days. The aim is to characterize their temporal fluctuations to support the identification of relevant biomarkers for clinical monitoring and risk stratification in relation to the pediatric population suffering from mild TBIs.

## 2. Materials and Methods

This observational, prospective, longitudinal, and analytical cohort study was conducted according to the Declaration of Helsinki and the General Health Law on Research for Health in Mexico. The study protocol (NIP 035/2015) was approved by the Research and Ethics Committees of the National Institute of Pediatrics (IRB00008065 and IRB00008064), both of which are registered with the U.S. Office for Human Research Protections (OHRP).

Patient Selection: Pediatric patients aged 1 to 18 years, diagnosed with mTBI, and without prior neurological conditions or other diseases requiring pharmacological treatment were included in this study if they had arrived at the National Institute of Pediatrics’ emergency department within 3 h post-injury and provided informed consent. The control group included healthy children who were recruited during the same time of day (morning) as the TBI population, had demographic characteristics similar to those of the pediatric patients with mTBIs, had been diagnosed as healthy children, and agreed to participate in this study through an informed consent letter. They also participated after giving informed consent. Cases involving incomplete medical records, changed diagnoses, discontinued treatments, preexisting neurological conditions, consumption of foods that modify plasma levels of amino acids, or withdrawn consent were excluded from this study.

Sample Collection and Processing: Blood samples (2 mL) were collected from mTBI patients at 3, 6, 12, and 24 h as well as at 7, 14, and 28 days post-injury. Peripheral venous access was employed, and phlebotomy was standardized to a maximum of two attempts, under topical anesthesia (Aztra Zeneca, Mexico City, Mexico).

Samples were gathered in EDTA-treated tubes (BD Vacutainer^®^ Becton Dickinson CTR Scientific, Mexico City, Mexico), maintained at 4 °C, and centrifuged, and the plasma was preserved at −70 °C until analysis.

Amino Acid Quantification (HPLC): Plasma concentrations of glutamate, aspartate, glutamine, and glycine were measured using high-performance liquid chromatography (HPLC) with fluorometric detection (Agilent Technologies mod. 1100, Santa Clara, CA, USA). Every 100 μL plasma aliquot was mixed with 0.1 M perchloric acid, neutralized with potassium carbonate, centrifugated (14,000 rpm form 10 minutes) and finally added with orthophthaldehyde (SIGMA, Missouri, USA) in equal parts before injection to induce fluorescence. The detection was conducted at 360 nm excitation and 450 nm emission wavelengths. A C18 reverse-phase column (Zorbax C18, Agilent Technologies, Santa Clara, CA, USA) 115 was employed under a sodium acetate-and-methanol gradient (buffer A and buffer B), starting with a flow rate of 0.5 mL/min at 25 °C.

The sample run time lasted 15 min. Amino acids were quantified using the external standard method, where peak areas are compared to calibration curves specific to each analyte.

Statistical Analysis: Descriptive statistics were utilized based on the type of variable: categorical data are presented as frequencies and percentages, while continuous data are shown as means ± standard deviations (SDs). Normality was assessed using the Kolmogorov–Smirnov test. Our primary analysis centered on the temporal variation in amino acid concentrations, employing paired *t*-tests for each time point in comparison to day 28, following a K−1 model. Differences in means, 95% confidence intervals (CIs), and *p*-values (with figures below 0.05 signifying significance) are reported.

Effect sizes were calculated using Cohen’s d and interpreted as follows: small (0.2–0.49), moderate (0.5–0.79), large (0.8–1.29), and very large (≥1.3). The assessment of effect size is crucial for evaluating the clinical relevance of biochemical alterations, guiding future research, and making clinical decisions. All analyses were conducted using SPSS version 22 (IBM Corp., Armonk, NY, USA).

## 3. Results

Data from 36 pediatric patients with mTBIs and 44 controls without TBIs were analyzed. This analysis incorporated various factors, including clinical and demographic variables, as well as plasma amino acid profiles. The profiles comprised concentrations of glutamate, aspartate, glutamine, and glycine; glutamine/glutamate ratios; and the GLX index (glutamate + glutamine). There was a male predominance in both groups. The most common locations where TBI-related accidents occurred included the patient’s home (52.78%), the street (27.7%), and school (11.11%). Falls were identified as the predominant mechanism of injury (80.56%), followed by bicycle accidents (13.89%) and blunt head trauma due to objects (2.78%). For a thorough overview of the demographic characteristics, refer to Table 1.

Among pediatric patients with mTBIs, the most commonly observed clinical variables included abnormal findings on cranial CT for 53.6% of the 28 patients who were subjected to this imaging technique; none of these instances necessitated surgical intervention or changed the initial diagnosis determined using the GCS. In addition, 75% of the patients exhibited subgaleal hematomas, and 55.6% reported instances of vomiting. Detailed results regarding these clinical variables are presented in Table 2.

### 3.1. Amino Acid Profiles of Pediatric Patients with mTBIs

We analyzed plasma concentrations of glutamate, aspartate, glutamine, and glycine in pediatric patients with mTBIs at multiple intervals post-injury (3 h, 6 h, and 12 h and 1, 7, 14, and 28 days afterwards). These measurements were compared to those for the control group. Table 3 presents the mean concentrations ($\bar{x}$), SDs, and 95% CIs in micromolar (µM), providing a detailed overview of the changes in amino acid levels over time following the traumatic event.

Following the TBI, there was an initial decrease in plasma glutamate levels, which dropped to 76.5% of that for the control group at 3 h post-injury. From 6 h to day 1, these levels remained relatively stable. However, a significant increase was observed at 7 days, with levels rising to 132.9%. After this peak, glutamate levels gradually declined, ultimately stabilizing at 81.8% by day 28 (Table 3).

Plasma levels of aspartate significantly decreased, reaching 83.2% of the levels for the control group 3 h post-TBI. Although modest recovery occurred in the following hours, the concentrations remained below the control values. Fourteen days post-injury, levels notably increased to 131.9% but then declined to 97.3% by day 28, approaching baseline levels (Table 3).

Following the TBI, plasma glutamine concentrations progressively increased, starting from 3 h post-injury. They initially rose to 170.6% relative to the control group and continued to climb overtime, reaching a peak value of 288% at 28 days post-trauma (Table 3).

An initial decrease in plasma glycine levels was observed, with levels dropping to 66.2% of those in the control group at 3 h post-TBI. This reduction persisted throughout the first 12 h. From day 7 post-injury onward, consistent and progressive recovery occurred, culminating in an average concentration of 127.8% by day 28 (Table 3).

### 3.2. Analysis of the Glutamine/Glutamate Index and the GLX Index (Glutamate + Glutamine) in Patients with mTBIs

In the TBI patient group, the glutamine/glutamate ratio increased by 291.7% at 3 h post-injury relative to the control group. This upward trend continued until day 1, reaching 361.3%, primarily driven by elevated levels of both amino acids, with a more pronounced increase in glutamine. However, by day 7, the ratio showed a relative decline due to a rise in glutamate levels. From day 14 onward, the ratio began to increase again, peaking at 442.7% by day 28 (Table 4).

In the TBI group, the GLX index increased to 180.7% at 3 h post-injury relative to the control group, indicating an early accumulation of glutamate and glutamine. This elevated level persisted until day 14, reaching 258.7%, and then slightly stabilized at 263.1% by day 28 (Table 4).

### 3.3. Amino Acid Kinetics over Time in Patients with mTBIs: Analysis of the Magnitude of Change in Pairwise Comparison with the 28-Day Post-Injury Baseline

On day 7, glutamate levels exhibited a brief spike, marked by a statistically significant difference (*p* = 0.030) and a moderate-to-high effect size (d = 0.70). The negative mean difference (−13.37 µM) indicates that glutamate concentrations were considerably higher on day 7 in comparison to day 28. Although the difference on day 14 was not statistically significant, the moderate effect size (d = 0.47) suggests persistently high plasma levels (Figure 1A).

In the case of aspartate, no statistically significant differences were observed at the early post-injury time points (3 h to 7 days), with effect sizes ranging from −0.38 to 0.31. These results suggest there were no clinically relevant changes. However, by day 14, a trend towards a moderate effect was noted (d = 0.62; *p* = 0.103), indicating that, by day 28, aspartate levels decreased, approaching the baseline concentrations (Figure 1B).

Glutamine levels were significantly lower during the initial 3 h up to day 7 compared to day 28 (*p* < 0.001 for all comparisons). The effect sizes ranged from large to very large (d = −0.81 to −1.27), indicating a steady and consistent increase over time. By day 14, however, no significant differences were observed (*p* = 0.284, d = −0.01), as the concentrations approached those observed on day 28 (Figure 1C).

Glycine concentrations consistently increased at all the measured time points. Between 3 and 12 h post-injury, effect sizes ranged from −1.08 to −1.44, indicating a very large effect. Although the effect size decreased slightly by 24 h, it remained significant (d = −0.79; *p* = 0.016). By day 7, the effect size was further reduced to a moderate level (d = −0.53; *p* = 0.149). The highest glycine concentrations were observed on the 14th and 28th days (Figure 1D).

Regarding the GLN/GLU ratio, significant variations were observed, showing an overall upward trend. Statistically significant differences were noted compared to day 28 during the initial 12 h, with small-to-moderate effect sizes driven by increased glutamine levels. A decrease in the ratio occurred on day 7 (d = −0.44; *p* = 0.064), coinciding with the peak in glutamate levels and the initial dramatic rise in glutamine levels (Figure 1E).

The GLX index, measuring the combined concentration of glutamate and glutamine, exhibited a high and statistically significant effect size within the first 12 h post-injury. By day 7, the effect size diminished to a moderate level (d = −0.68; *p* = 0.003). These findings indicate a progressive and clinically meaningful increase in the levels of both amino acids, with statistical significance persisting until day 7. Thereafter, no significant differences were detected (d = −0.06; *p* = 0.318) (Figure 1F).

## 4. Discussion

The demographic and clinical characteristics of the pediatric population with mTBIs in our study align with those observed in other international cohorts: there is a male predominance, falls are the most common cause of injury, and the most frequent clinical manifestations are headaches, vomiting, altered consciousness levels, temporary mental status changes like confusion or disorientation, and transient neurological deficits like focal signs or seizures [1,3].

This study presents evidence of dynamic and longitudinal changes in the plasma profiles of excitatory and inhibitory amino acids (glutamate, glutamine, glycine, and aspartate), along with the GLN/GLU (glutamine/glutamate) and GLX (the aggregate of glutamate and glutamine) indices. These parameters are vital for assessing glutamatergic metabolic stability as they illustrate the equilibrium between excitotoxicity—mainly mediated by glutamate—and astrocytic regulatory mechanisms accountable for glutamate’s conversion into glutamine. Our discoveries enhance our understanding of the neurochemical pathophysiology of TBIs and provide support for the prospective use of these substances as diagnostic, monitoring, and prognostic biomarkers.

Glutamate, the primary excitatory neurotransmitter in the central nervous system, plays a crucial role in synaptogenesis, neuronal plasticity, memory, and learning. It is widely associated with excitotoxic mechanisms in TBIs [63,64]. In our pediatric cohort, plasma glutamate levels initially fell and then rose steadily to a significant peak on day 7. Preclinical studies have reported a swift surge in extracellular cerebral glutamate post-injury; this surge is tied to intense neuronal discharges and potentially linked to long-term brain damage [65].

It is important to note that these findings originate from cerebral microdialysis studies obtained using moderate-to-severe TBI models, while our study centers on the quantification of peripheral plasma in pediatric mTBI patients. Thus, the observed increases could reflect secondary and time-dependent pathological processes. The gradual rise in glutamate levels may be due to impaired astrocytic reuptake mechanisms—governed by GLT-1 and GLAST—and increased BBB permeability. This would explain the significant rise observed during the subacute phase (day 7), a phenomenon previously tied to glial dysfunction and secondary inflammation in both animal models and human studies of moderate-to-severe TBIs or stroke [64,66,67,68,69]. Timofeev and colleagues documented persistently high extracellular glutamate levels during the first week following severe TBIs in patients with fatal outcomes [57]. In our cohort, the decrease observed on day 14 suggests a partial recovery of glutamatergic metabolism, whereas the sustained elevation on day 28 may signal prolonged dysfunction. These findings underscore glutamate’s potential as a subacute biomarker of post-traumatic neurochemical disruption. Glutamine, often viewed as a neurochemical buffer against excessive extracellular glutamate levels, serves as a metabolic reservoir. Astrocytic synthesis of glutamine is pivotal in the mitigation of excitotoxicity and the prevention of progressive neuronal damage [64,70]. The relationship between these amino acids can be evaluated using the GLN/GLU and GLX indices, which reflect the astrocytic conversion of glutamate into glutamine and the overall metabolic statuses of excitatory substances, respectively [71,72]. In our study sample, glutamine levels progressively and consistently increased throughout the study. This finding could be linked with the surge in intracerebral glutamate levels detailed by Katayama et al. and the peak recorded in our study on the seventh day. Initially, the GLN/GLU index showed glutamine predominance, with a temporary inversion on day seven; this coincided with a rise in plasma glutamate levels, possibly resulting from glial neuroinflammation or energy reprogramming instigated by the TBI [65,73,74]. The following sustained increase in glutamine levels may indicate a glial detoxification strategy [64]. This notion is corroborated by studies that show increased glutamine synthesis in response to astrocyte activation [75]. The continued rise in the GLN/GLU index until day 28 might indicate ongoing glial activation or subclinical inflammation, potentially affecting synaptic plasticity and functional prognosis, as Richards and colleagues found in regard to pediatric TBI patients [76]. A steep decrease in this index may serve as an early warning of excitotoxicity, while sustained elevation might signal ongoing compensatory mechanisms. Thus, the GLX index—which integrates both excitotoxicity and glial responses—has been found to be a promising potential biomarker for early monitoring and functional risk stratification.

Glycine, an inhibitory neurotransmitter and NMDA receptor co-agonist, modulates neuronal excitability in the presence of glutamate [77]. In the acute phase, a significant elevation was observed, likely reflecting an inhibitory response to increased glutamate levels [65]. Previous work has shown that glycine modulates NMDA receptor channel opening, regulates calcium influx, and interferes with intracellular signaling cascades linked to cell death [78]. Its decrease from day 7 and stabilization by day 14 suggest there is a transient compensatory mechanism that counteracts heightened excitability [79].

Levels of aspartate, which participates in both excitatory neurotransmission and intracellular metabolic pathways, showed a slight decline during the acute phase. This decrease may act as a compensatory mechanism for avoiding excitotoxicity. This reaction was followed by a modest increase, though it did not meet the control levels, peaking on day 14 and then decreasing by day 28. Despite the lack of statistical significance, these variations could indicate post-traumatic neuronal plasticity or metabolic changes. Considering aspartate’s role in mitochondrial modulation after a brain injury [68,80,81], it might serve as more of a transitional biomarker as opposed to a primary one.

Finally, it is worth noting that the circulating amino acids analyzed in this study do not originate exclusively from the CNS, even in conditions where the BBB may be compromised. Plasma amino acid levels are influenced by multiple peripheral physiological processes, including intestinal absorption after food intake; muscle protein synthesis and degradation; hepatic metabolism, especially in the urea cycle and transamination; renal filtration and reabsorption; and, to a lesser extent, bidirectional transport regulated across the BBB by specific transporters.

These mechanisms allow for a relatively stable degree of plasma homeostasis, even in the presence of dietary variations or moderate fasting, as documented in previous studies [82]. Therefore, plasma concentrations of glutamate, glutamine, and glycine reflect not only brain metabolism but also the dynamic balance of these multiple compartments.

Nevertheless, the use of peripheral blood as a biological matrix was a methodological decision based on feasibility, clinical ethics, and the future applicability of biomarkers in real-life pediatric care settings. While we recognize that it does not directly discern the cerebral origin of amino acids, the longitudinal changes observed after a TBI reflect a systemic response that may be influenced by mechanisms occurring after the traumatic event. Furthermore, the control group—comprising healthy children without TBIs or neurological or metabolic diseases—was essential for establishing a representative baseline of plasma amino acid concentrations in the Mexican pediatric population. This group was evaluated under controlled conditions, including morning collection and without prolonged fasting, to minimize the effect of diurnal or dietary variations. It is worth mentioning that the specialized literature indicates that, in the absence of severe malnutrition or hepatorenal diseases, diet alone does not induce significant changes in plasma levels of individual amino acids, especially those such as glutamate and glutamine that have a tightly regulated metabolism [83].

## 5. Conclusions

Our findings indicate that mTBIs in members of the pediatric population correlate with dynamic and sustained changes in plasma amino acid levels, particularly glutamate, glutamine, and glycine. These changes signal ongoing neurochemical processes involved in excitotoxicity, astrocytic regulation, and secondary inflammation. The chronological progression of these profiles, along with the GLN/GLU and GLX indices, provides a comprehensive view of post-traumatic glutamatergic metabolism and underscores their potential as diagnostic and prognostic tools.

This study is one of the first to confirm the biological reliability of using plasma amino acids as biomarkers for pediatric mTBIs, thereby setting the stage for the development of more sensitive, specific, and personalized monitoring strategies. However, further research involving larger cohorts is required in order to validate these findings and explore their connection with long-term clinical and neurocognitive outcomes. Incorporating these biomarkers into pediatric clinical practice and comparing them with traditional biomarkers, such as NSE or SB100, could enable earlier detection of neurochemical dysfunction and expedite therapeutic interventions.

## Figures and Tables

**Figure 1 neurolint-17-00145-f001:**
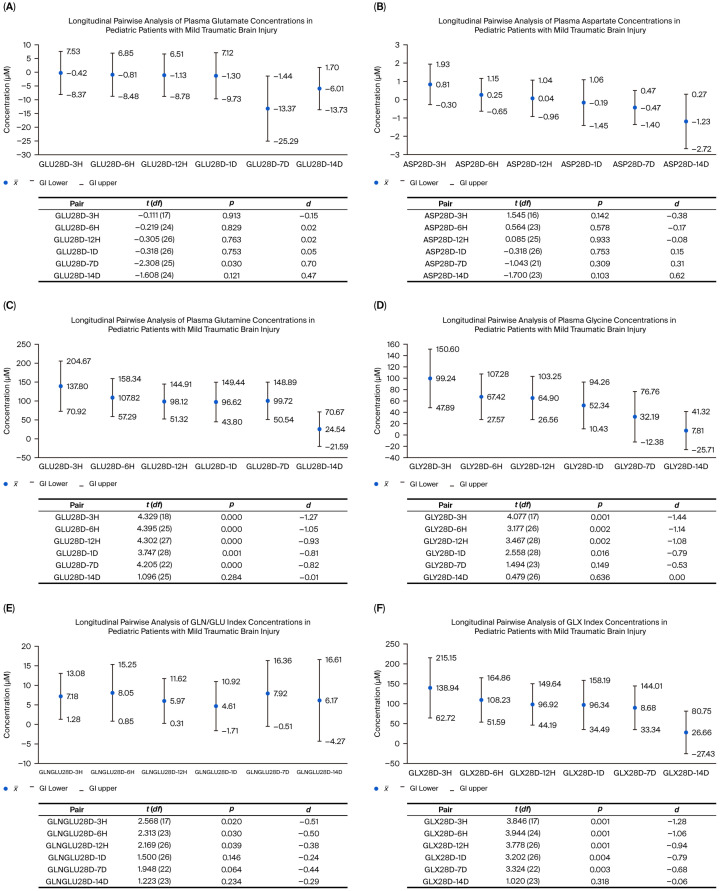
Kinetics of plasma amino acids over time in pediatric patients with mTBIs. Each point was compared to the 28-day value to determine the magnitude of change (Cohen’s *d*). The *Y*-axis shows concentration differences (µM) with 95% confidence intervals (CIs). Negative values indicate higher concentrations on day 28; positive values indicate reductions on day 28 relative to earlier time points.

**Table 1 neurolint-17-00145-t001:** Demographic and anthropometric characteristics. The percentages of the different demographic and anthropometric characteristics of the two populations analyzed are shown: the controls, or those without a TBI, with 44 individuals representing 100%, and the population of patients with mild TBIs, with 36 individuals representing 100%.

Variable	Without TBI (*n* = 44) %	Mild TBI (*n* = 36) %
Gender		
Male/Female	54.5/45.5	69.4/30.6
Age (years)		
0–2 (neonate and infant)	6.8	33.3
3–5 (preschool)	11.4	27.8
6–11 (school-age)	45.4	27.8
12–18 (adolescent)	36.4	11.1
Body Mass Index (kg/m^2^)		
Underweight	13.6	20.7
Normal	70.5	58.6
Overweight	11.4	6.9
Obesity	4.5	13.8

**Table 2 neurolint-17-00145-t002:** Clinical variables of patients with mild TBI.

Clinical Variable	Frequency (%)
Abnormal CT scan	28 (53.6%)
Post-traumatic headache	27 (75%)
Subgaleal hematoma	27 (75%)
Vomiting	20 (55.6%)
Loss of alertness	8 (22.2%)
Mental alteration	6 (16.7%)
Seizures	4 (11.1%)
Post-traumatic amnesia	2 (5.6%)
Abnormal neurological examination	2 (5.6%)

**Table 3 neurolint-17-00145-t003:** Evolution of plasma levels of amino acids in pediatric patients over time. The $\bar{x}$ ± SD of the plasma concentrations (in µM) of the different amino acids analyzed are shown as well as the 95% confidence interval at the various times analyzed after the mTBI.

Amino Acids	CONTROL $\bar{x}$ (µM) ± SD (95% CI)	mTBI 3 h $\bar{x}$ (µM) ± SD (95% CI)	mTBI 6 h $\bar{x}$ (µM) ± SD (95% CI)	mTBI 12 h $\bar{x}$ (µM) ± SD (95% CI)	mTBI 1 D $\bar{x}$ (µM) ± SD (95% CI)	mTBI 7 D $\bar{x}$ (µM) ± SD (95% CI)	mTBI 14 D $\bar{x}$ (µM) ± SD (95% CI)	mTBI 28 D $\bar{x}$ (µM) ± SD (95% CI)
Glutamate	30.81 ± 9.57 (27.8–33.8)	23.56 ± 9.6 (19.7–27.4)	25.44 ± 10.91 (21.5–29.4)	25.44 ± 11.89 (21.3–29.6)	25.76 ± 12.52 (21.4–30.1)	40.94 ± 28.63 (30.4–51.5)	32.15 ± 16.8 (25.6–38.7)	25.19 ± 12.28 (20.4–29.9)
Aspartate	4.52 ± 0.68 (4.3–4.7)	3.76 ± 1.71 (3.1–4.5)	4.13 ± 1.49 (3.6–4.7)	4.27 ± 1.61 (3.7–4.8)	4.71 ± 2.39 (3.9–5.5)	4.97 ± 2.12 (4.1–5.8)	5.96 ± 3.15 (4.7–7.2)	4.40 ± 1.63 (3.8–5.0)
Glutamine	142.82 ± 34.49 (131.5–154.2)	278.77 ± 87.42 (243.6–314.1)	304.71 ± 84.11 (274.9–335.0)	310.34 ± 99.97 (275.5–345.2)	322.93 ± 102.04 (287.9–358.0)	325.36 ± 89.83 (289.8–360.9)	410.66 ± 135.02 (359.3–462.0)	411.83 ± 118.36 (366.8–456.9)
Glycine	152.82 ± 57.9 (134.8–170.8)	101.13 ± 20.57 (92.4–109.8)	121.58 ± 34.26 (109.2–133.9)	127.61 ± 30.62 (116.9–138.3)	145.34 ± 33.18 (133.8–156.9)	159.19 ± 42.85 (142.6–175.8)	194.91 ± 76.26 (166.4–223.4)	195.25 ± 85.88 (163.2–227.3)

**Table 4 neurolint-17-00145-t004:** Evolution of the glutamine/glutamate ratio and GLX ratio in the plasma of the pediatric patients studied. The $\bar{x}$
+ SD of the glutamine/glutamate ratio and the GLX index are shown as well as the 95% confidence interval at the different times analyzed after the mTBI.

Study Indices	CONTROL $\bar{x}$ (µM) ± SD (95% CI)	mTBI 3 h $\bar{x}$ (µM) ± SD (95% CI)	mTBI 6 h $\bar{x}$ (µM) ± SD (95% CI)	mTBI 12 h $\bar{x}$ (µM) ± SD (95% CI)	mTBI 1 D 24 h $\bar{x}$ (µM) ± SD (95% CI)	mTBI 7 D $\bar{x}$ (µM) ± SD (95% CI)	mTBI 14 D $\bar{x}$ (µM) ± SD (95% CI)	mTBI 28 D $\bar{x}$ (µM) ± SD (95% CI)
Glutamine/Glutamate Ratio	5.06 ± 1.68 (4.5–5.6)	14.76 ± 8.95 (11.1–18.4)	15.26 ± 8.69 (12.1–18.5)	16.39 ± 12.86 (11.9–20.9)	18.28 ± 15.41 (12.9–23.7)	14.88 ± 14.96 (9.0–20.8)	17.36 ± 15.16 (11.7–23.4)	22.4 ± 18.77 (15.1–29.7)
GLX Index (Glutamate + Glutamine)	167.31 ± 35.25 (155.4–179.2)	302.33 ± 88.20 (266.7–338)	330.15 ± 83.59 (300–360.3)	335.78 ± 99.81 (301–370.6)	350.73 ± 104.19 (314.4–387.1)	363.21 ± 102.4 (322.7–403.7)	432.8 ± 143.63 (376–489.6)	440.16 ± 122.49 (392.7–487.7)

## Data Availability

The datasets supporting the conclusions of this manuscript are not publicly available in order to ensure data confidentiality. Requests to access the datasets should be directed to aperez@facmed.unam.mx.

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
