# Peer review of "Biological Plausibility of Using Plasma Amino Acid Profile Determination as a Potential Biomarker for Pediatric Patients with Mild Traumatic Brain Injuries"

_2035-8377, 2025, doi:10.3390/neurolint17090145_

Round 1
Reviewer 1 Report
Comments and Suggestions for Authors
The authors have shown neurological assessment of mTBI using amino acid profiling (as biomarkers)
I have a few concerns that may require revisions
1) Table 1: Male 54.5 and 69.4 in control and mTBI, respectively, are these values percentages? They do not add up to 100%
2) Tables need a bit more concise legends to explain (not just the Table title)
3) Shouldn't there be control time courses in some of the Tables?
4) The unit hour is indicated as h in the text but H in the Table. Should be more consistent.
5) In some Table it was IC95% but in some 95% CI? In the boxes, inside the parenthesis, is it a range , or IC95%?
6) authors can also explain a bit more on the sources of amino acid possibly not from the leaky BBB from brain, but from other organ systems since peripheral blood was collected.
Author Response
The authors have shown neurological assessment of mTBI using amino acid profiling (as biomarkers) I have a few concerns that may require revisions
1) Table 1: Male 54.5 and 69.4 in control and mTBI, respectively, are these values percentages? They do not add up to 100%
ANSWER:
We appreciate your observation. Thank you for your comment. In Table 1, each column corresponds to a different population. In the case of the population without TBI, the percentage of males is 54.5%, and that of females is 45.5%, totaling 100%. For the population with TBI, the percentage of men is 69.4% and that of women is 30.6%. We added the female percentage to the table.
In global statistics on the incidence of head trauma, the male population suffers the most. It is worth noting that the same thing happens in our population.
2) Tables need a bit more concise legends to explain (not just the Table title)
ANSWER:
Your comment is appreciated and to facilitate the reading of your observation, legends have been added to each of the tables used in the manuscript.
3) Shouldn't there be control time courses in some of the Tables?
ANSWER:
We appreciate your comment, which allows us to delve deeper into an important methodological aspect of the study. Our tables do not show time courses for the control group, reflecting the intentional design of sample collection in this cohort.
The control group consisted of a healthy pediatric population, carefully selected according to strict clinical criteria: no history of traumatic brain injury (TBI), no prior medical conditions, and no exposure to pharmacological treatments that could alter amino acid metabolism. Due to ethical and logistical considerations—specific to studies in healthy pediatric populations—only a single blood sample was taken, at a consistent morning schedule, and under conditions comparable to those of the TBI group at admission.
It should be noted that this control group was not designed as a cohort with longitudinal follow-up, as the Ethics Committee determined that performing multiple samples in healthy children would represent an unnecessary burden for this population. Therefore, the control group in our study serves as a baseline reference point against which amino acid concentrations are compared at different time points after the TBI in the affected cohort.
Given that there is very little standardized data on baseline amino acid levels in the pediatric population in the literature, and even less so in the Mexican population, we believe that this group plays a key role by providing a physiological reference framework, especially useful in interpreting the biochemical changes observed after trauma. As has been reported in some international studies, reference values ​​can vary significantly with age.
4) The unit hour is indicated as h in the text but H in the Table. Should be more consistent.
ANSWER:
We appreciate your comment. The abbreviations have been reviewed and standardized throughout the text.
5) In some Table it was IC95% but in some 95% CI? In the boxes, inside the parenthesis, is it a range, or IC95%?
ANSWER:
We appreciate your comment and to address this observation, the manuscript was revised and the confidence interval terminology was standardized in every table or statement used.
6) Authors can also explain a bit more on the sources of amino acid possibly not from the leaky BBB from brain, but from other organ systems since peripheral blood was collected.
ANSWER:
We appreciate your observation, which allows us to delve deeper into the systemic origin of plasma amino acids and their interpretation in the context of mild traumatic brain injury (TBI) in the pediatric population.
Circulating amino acids in peripheral blood do not originate exclusively from the central nervous system (CNS), even in conditions where the blood-brain barrier (BBB) ​​may be compromised. Plasma amino acid levels are influenced by multiple peripheral physiological processes, including intestinal absorption after food intake, muscle protein synthesis and degradation, hepatic metabolism, especially in the urea cycle and transamination, renal filtration and reabsorption, and, to a lesser extent, bidirectional transport regulated across the BBB by specific transporters.
These mechanisms allow for a relatively stable degree of plasma homeostasis, even in the presence of dietary variations or moderate fasting, as documented in previous studies (Cynober et al., 2002). Therefore, plasma concentrations of glutamate, glutamine, and glycine reflect not only brain metabolism, but also the dynamic balance of these multiple compartments.
The use of peripheral blood as a biological matrix was a methodological decision based on feasibility, clinical ethics, and the future applicability of biomarkers in real-life pediatric care settings. While we recognize that it does not directly discriminate the cerebral origin of amino acids, we believe that the longitudinal changes observed after TBI reflect a systemic response that may be influenced by mechanisms following the traumatic event. For this reason, the control group—comprised of healthy children without TBI or neurological or metabolic diseases—was essential to establish a representative baseline of plasma amino acid concentrations in the Mexican pediatric population. This group was evaluated under controlled conditions, including morning collection and without prolonged fasting, to minimize the effect of diurnal or dietary variations. It is worth mentioning that specialized literature indicates that, in the absence of severe malnutrition or hepatorenal diseases, diet alone does not induce significant changes in plasma levels of individual amino acids, especially those such as glutamate and glutamine that have a tightly regulated metabolism (Reeds et al., 2000).
Cynober LA. Plasma amino acid levels with a note on membrane transport: characteristics, regulation, and metabolic significance. Nutrition. 2002 Sep;18(9):761-6. doi: 10.1016/s0899-9007(02)00780-3. PMID: 12297216.
Reeds PJ, Burrin DG, Stoll B, Jahoor F. Intestinal glutamate metabolism. J Nutr. 2000 Apr;130(4S Suppl):978S-82S. doi: 10.1093/jn/130.4.978S. PMID: 10736365. We added a paragraph in the Discussion Section lines 404-428.
Reviewer 2 Report
Comments and Suggestions for Authors
The manuscript presents a prospective longitudinal cohort study analyzing plasma concentrations of glutamate, aspartate, glutamine, and glycine — along with GLN/GLU and GLX indices — in pediatric patients with mild TBI over 28 days, comparing them with healthy controls. The results show dynamic neurochemical changes that may serve as potential biomarkers for diagnosis and monitoring.
The work addresses a clinically relevant and understudied area: biochemical biomarkers in pediatric mTBI. While amino acids have been explored in adult or severe TBI, their longitudinal profile in children with mild injury is novel.
Their Findings could contribute to developing non-invasive monitoring tools, potentially impacting early diagnosis and prognosis in pediatric neurotrauma.The study design (prospective, longitudinal) is appropriate. The use of repeated sampling over multiple time points adds robustness. Statistical analysis is adequately described, including effect sizes. However, sample size is relatively small, and potential confounders (diet, metabolic disorders, medications) are not deeply addressed.
The manuscript is generally well-written, logically structured, and free of major language errors. Figures and tables are clear. The introduction provides comprehensive background. Some discussion sections could be condensed to improve focus.
Introduction May benefit from a more concise focus on knowledge gaps specific to pediatric mTBI.
Consider addressing potential influences of diet or fasting state on plasma amino acid levels in M&M
Recommendation - Minor Revision
Justification: The study is novel, methodologically sound, and relevant. Minor improvements in presentation, additional acknowledgment of confounders, and discussion refinement would strengthen the manuscript before acceptance.
Author Response
The manuscript presents a prospective longitudinal cohort study analyzing plasma concentrations of glutamate, aspartate, glutamine, and glycine — along with GLN/GLU and GLX indices —in pediatric patients with mild TBI over 28 days, comparing them with healthy controls. The results show dynamic neurochemical changes that may serve as potential biomarkers for diagnosis and monitoring.
The work addresses a clinically relevant and understudied area: biochemical biomarkers in pediatric mTBI. While amino acids have been explored in adult or severe TBI, their longitudinal profile in children with mild injury is novel.
Their Findings could contribute to developing non-invasive monitoring tools, potentially impacting early diagnosis and prognosis in pediatric neurotrauma. The study design (prospective, longitudinal) is appropriate. The use of repeated sampling over multiple time points adds robustness. Statistical analysis is adequately described, including effect sizes. However, sample size is relatively small, and potential confounders (diet, metabolic disorders, medications) are not deeply addressed.
The manuscript is generally well-written, logically structured, and free of major language errors. Figures and tables are clear. The introduction provides comprehensive background. Some discussion sections could be condensed to improve focus.
Introduction May benefit from a more concise focus on knowledge gaps specific to pediatric mTBI.
Consider addressing potential influences of diet or fasting state on plasma amino acid levels in M&M
Recommendation - Minor Revision
Justification: The study is novel, methodologically sound, and relevant. Minor improvements in presentation, additional acknowledgment of confounders, and discussion refinement would strengthen the manuscript before acceptance.
ANSWER:
We appreciate your comments; the document has been modified in the requested sections to improve its readability. See Introduction section, lines 87-88, 97-103, 112-113; Materials and Methods section, lines 130-140 and Discussion section, lines 404-428.
Reviewer 3 Report
Comments and Suggestions for Authors
summary:
The manuscript explores whether plasma amino-acid trajectories (glutamate, aspartate, glutamine, glycine) and derived indices (GLN/GLU and GLX) can serve as biomarkers in pediatric mild TBI (mTBI) over 28 days. The study is prospective and longitudinal with serial sampling in 36 children with mTBI and 44 controls, and uses HPLC with fluorometric detection for quantification, followed by hypothesis-testing against the day-28 value and effect-size reporting.
strengths:
The longitudinal design with dense early sampling (3, 6, 12, 24 h; 7, 14, 28 d) is a major asset for capturing acute and subacute biochemical dynamics in children—an understudied population in whom adult biomarkers are not yet validated for routine use. The HPLC method (OPA derivatization; FL-45 detector; C18 gradient; 3.5 mL/min at 35 °C; 15-min run-time) is standard and clearly described, lending credibility to the measurements.
weaknesses:
- Stats: The choice of day 28 as the reference for paired t-tests (“K-1 model”) is unconventional and risks misinterpretation: day 28 is not a true pre-injury baseline, and in pediatric mTBI there can be ongoing pathophysiology at this time. A trajectory-based model that uses all time points jointly—and compares mTBI vs controls via a group×time interaction—would be more appropriate (e.g., linear mixed-effects with subject random intercepts).
- -Clinical profiles: The cohort includes many very young children (0–2 y: 33.3% of mTBI), and over half of those imaged had CT abnormalities (53.6%), consistent with “complicated” mTBI in a substantial fraction. These features can influence peripheral amino-acid levels and recovery kinetics. Age-stratified analyses (e.g., 0–5, 6–11, 12–18) and reporting by CT-positive vs CT-negative (“complicated” vs “uncomplicated” mTBI) would sharpen inference. Provide a table or supplement that details the types of CT lesions (e.g., contusion, SAH, EDH/SDH) and any interventions. (Age/CT data and clinical variables.
- preanalytical assessment: the meethods specify EDTA plasma, storage at −70 °C, and OPA derivatization with FL detection, which is appropriate; however, critical pre-analytical variables that affect glutamate/glycine (fasting status, time-of-day, hemolysis/RBC contamination checks, diet, medications/sedation, and time from draw to freeze) are not reported. Given amino-acid diurnal/feeding effects, these should be included
- Data interpretation: The manuscript highlights a day-7 glutamate peak with subsequent partial decline, progressive glutamine increases, an early GLX rise, and GLN/GLU dynamics. These descriptive findings are interesting, but causality is speculative without outcome linkage. Adding clinical outcomes (e.g., symptom scales, return-to-school, PPCS), or—even better—testing whether GLN/GLU or GLX predict outcomes after adjusting for age/sex/CT would greatly elevate impact.
- Back to point # 2, The study does not benchmark the proposed amino-acid measures (GLU, ASP, GLN, GLY; GLN/GLU; GLX) against established TBI protein biomarkers (e.g., GFAP and UCH-L1 for acute/astroglial–neuronal injury; S100B in pediatrics; NfL and tau for axonal/subacute injury). Without head-to-head comparisons, it is difficult to judge the incremental diagnostic or prognostic value of these metabolites or to position them within existing clinical pathways.
Several typos and grammar mistakes
Author Response
The manuscript explores whether plasma amino-acid trajectories (glutamate, aspartate, glutamine, glycine) and derived índices (GLN/GLU and GLX) can serve as biomarkers in pediatric mild TBI (mTBI) over 28 days. The study is prospective and longitudinal with serial sampling in 36 children with mTBI and 44 controls, and uses HPLC with fluorometric detection for quantification, followed by hypothesis-testing against the day-28 value and effect-size reporting.
Strengths:
The longitudinal design with dense early sampling (3, 6, 12, 24 h;7, 14, 28 d) is a major asset for capturing acute and subacute biochemical dynamics in children—an under studied population in whom adult biomarkers are not yet validated for routine use. The HPLC method (OPA derivatization; FL-45 detector; C18 gradient;3.5 mL/min at 35 °C; 15-min run-time) is standard and clearly described, lending credibility to the measurements.
Seaknesses:
- Stats: The choice of day 28 as the reference for paired t-tests (“K-1 model”) is unconventional and risks misinterpretation: day 28 is not a true pre-injury baseline, and in pediatric mTBI there can be ongoing pathophysiology at this time. A trajectory-based model that uses all time points jointly—and compares mTBI vs controls via a group×time interaction—would be more appropriate (e.g., linear mixed-effects with subject random intercepts).
ANSWER:
We appreciate the comment regarding the use of day 28 as the reference point for paired t tests in the K-1 model. We agree that, from a methodological perspective, this choice may seem unconventional as it does not represent a true pre-injury baseline value. However, we clarify that this decision was based on a series of clinical, ethical, and methodological considerations.
Day 28 post-trauma was selected as the comparison point because it represents a clinically relevant time point in the course of pediatric mild traumatic brain injury (TBI). Based on clinical experience, this time point was identified as one at which changes resulting from the traumatic event can be most clearly observed, allowing for a more stable assessment of plasma amino acid concentrations in relation to post-traumatic effects.
The control group was evaluated at a single time point, as it was a healthy pediatric population selected according to strict diagnostic criteria: no neurological history, no use of medications that alter metabolism, and no known metabolic conditions. The study design and ethical approval restricted the possibility of longitudinal follow-up in this population, considering that repeated exposures (such as blood draws) could represent unnecessary suffering, a key criterion in pediatric studies.
From this perspective, the single sample from a healthy population was considered a valid representation of the pre-trauma baseline state, as it was free of conditions that could alter plasma concentrations of the amino acids under study. Furthermore, all samples were taken in the morning, which reduced circadian metabolic variability and increased data homogeneity. To make this point clearer, the corresponding section was modified, see Materials and Methods section, line 130-140.
It is important to note that, although the analysis presented results centered on day 28 as the reference, all time points in the study were evaluated. The patterns observed in the trajectories of plasma amino acid concentrations were consistent regardless of the comparison point; that is, the trend of change attributable to the traumatic event was clear at all times studied. Selecting day 28 did not alter the fundamental findings but allowed for a more clinical and comparative interpretation of the persistent effects of mild TBI.
We recognize that a longitudinal trajectory model with a group × time interaction could offer a more robust perspective from a statistical analysis, and we agree with its potential for future research. However, in the present study, the methodological choice sought to balance clinical rigor, ethical feasibility, and interpretive relevance, without compromising the validity of the findings.
- Clinical profiles: The cohort includes many very young children (0–2 y: 33.3% of mTBI), and over half of those imaged had CT abnormalities (53.6%), consistent with “complicated” mTBI in asubstantial fraction. These features can influence peripheral amino-acid levels and recovery kinetics. Age-stratified analyses (e.g., 0–5, 6–11, 12–18) and reporting by CT-positive vs CT-negative (“complicated” vs “uncomplicated” mTBI) would sharpen inference. Provide a table or supplement that details the types of CT lesions (e.g., contusion, SAH, EDH/SDH) and any interventions. (Age/CT data and clinical variables.
ANSWER:
We sincerely appreciate your valuable comment, which we recognize as a contribution that would enrich the analytical and clinical scope of the study. However, in the current context of this first communication of results, we must clarify that it was not feasible to perform stratified analyses by age group or by subtypes of mild TBI based on CT findings due to several methodological and logistical limitations, which we explain below.
First, as we point out in the manuscript, the clinical utility of amino acids as biomarkers in mild TBI in the pediatric population is still in an exploratory phase, and the epidemiological design—although robust given its observational cohort—introduces certain systematic and random biases that directly impact the feasibility of performing more complex analyses.
The cohort includes a significant proportion of children aged 0–2 years (33.3% of the mild TBI group). While reflecting the epidemiological reality in the hospital setting, this limits the possibility of segmenting by broad age ranges (0–5, 6–11, 12–18 years) without compromising statistical power. Any additional stratification would have generated unstable estimates, with wide confidence intervals and a high risk of random error and spurious conclusions. 53.6% of patients with mild TBI presented changes consistent with "complicated" TBI, as part of routine hospital clinical practice. However, given the limited size of the cohort, a balanced distribution between subgroups (positive vs. negative for the presence of abnormalities found by the computed tomography study) was not achieved, allowing for stratified comparisons without loss of inferential validity.
Clinical data collection and sample collection strictly adhered to the time frames established by the Pediatric Emergency Department (3, 6, 12, 24 hours; 7, 14, and 28 days), which made it difficult to incorporate additional variables such as nutritional status, neurological development, social, or educational factors—aspects that could undoubtedly function as effect modifiers but are not part of the study that led to the findings presented.
Patient prognosis and outcome were observed only during the emergency room stay and immediate subsequent consultations. We lack extensive longitudinal follow-up that would allow for a differentiated assessment of recovery kinetics by age or injury type. Although bivariate analyses based on clinical and laboratory variables were explored, the sample limitations described made it impossible to develop multivariate models without incurring significant risks of overfitting, collinearity, and loss of internal validity. Based on these elements, we do not consider it methodologically appropriate at this stage to incorporate adjustments for confounding or effect-modifying variables. The available references on confounding control for amino acids in the context of TBI come mostly from studies in the adult population, and we do not have comparable literature to support such adjustments in the pediatric population, especially in children under 5 years of age.
While we recognize that including a supplement with detailed classification of CT findings (e.g., contusions, subarachnoid hemorrhages, subdural/epidural hematomas) and the need for interventions would have been valuable, this breakdown was not part of the original analysis and is therefore not included in the present study. However, we will consider your suggestion for future publications derived from the cohort, in which this clinical characterization can be further explored.
We are interest in conducting stratified analyses by age and CT injury type. However, given the exploratory phase of the study, sampling and ethical restrictions, and the lack of robust pediatric comparative frameworks, we chose to limit the analytical scope so as not to compromise the validity of the presented results. These aspects will be considered in future research.
- Preanalytical assessment: the methods specify EDTA plasma, storage at −70 °C, and OPA derivatization with FL detection, which is appropriate; however, critical pre-analytical variables that affect glutamate/glycine (fasting status, time-of-day, hemolysis/RBC contamination checks, diet, medications/sedation, and time from draw to freeze) are not reported. Given amino-acid diurnal/feeding effects, these should be included
ANSWER:
We appreciate your comment, based on which we clarify. The study design focused on a pediatric population with no relevant medical history: no previous metabolic, neurological, or systemic diseases, or chronic medication use. This criterion was established precisely to minimize the influence of pre-analytical confounding factors related to medications, comorbidities, or underlying metabolic disorders.
Although a strict fasting status was not imposed due to the nature of the hospital environment and the clinical needs of the pediatric emergency department, an effort was made to standardize the sample collection schedule, prioritizing morning sampling to reduce circadian variability in plasma amino acids. This measure sought to mitigate the diurnal and postprandial effects reported in the literature. The control group (without TBI) was also recruited under these same conditions, allowing for a comparative baseline analysis under the same sampling context, rather than under prolonged fasting conditions.
Standardized measures were implemented to prevent hemolysis, as this can significantly alter the concentrations of glutamate, glycine, and other intracellular amino acids. Blood collection was performed by trained clinical personnel using venous access devices that remained in place for the first 24 hours after the start of the collection protocol, avoiding multiple punctures. In addition, each blood sample was visually inspected for signs of hemolysis, and those showing any evidence of blood contamination were discarded.
Once the blood samples were obtained, they were immediately processed for plasma separation, which was stored at –70°C. The maximum time between extraction and freezing did not exceed 12 hours, according to the pre-established protocol. OPA derivatization and HPLC quantification were performed within a period of no more than one month from initial storage, complying with the stability standards reported for these compounds.
While we recognize that pre-analytical variables such as fasting status or diet can influence the plasma concentrations of certain amino acids, our protocol sought to control and standardize the most relevant factors within the real-world clinical context and the limitations of the pediatric study. Consistent collection time, rapid plasma separation, proper storage, and the exclusion of patients with confounding medical conditions reinforce the validity of the results presented.
- Data interpretation: The manuscript highlights a day-7glutamate peak with subsequent partial decline, progressive glutamine increases, an early GLX rise, and GLN/GLU dynamics. These descriptive findings are interesting, but causality is speculative without outcome linkage. Adding clinical outcomes (e.g., symptom scales, return-to-school, PPCS), or—even better—testing whether GLN/GLU or GLX predict outcomes after adjusting for age/sex/CT would greatly elevate impact.
ANSWER:
We appreciate your comment, which we consider pertinent to strengthen the clinical relevance of the reported biochemical findings. We agree that the value of biomarkers, such as glutamate (GLU), glutamine (GLN), GLX, and the GLN/GLU ratio, is significantly increased when they can be linked to objective clinical outcomes in patients with mild traumatic brain injury (TBI).
Following your recommendation, we conducted an exploratory multivariate analysis including variables such as age, sex, and computed tomography (CT) findings. However, we must note that the current sample size and subgroup segmentation made a robust statistical analysis impossible, as the number of cases per category was insufficient to obtain stable estimates. This situation was even more pronounced when attempting to link amino acid concentrations with clinical outcomes such as school return or persistent post-concussion symptoms. Furthermore, the uneven distribution between the control and TBI groups, as well as the lack of longitudinal data in the healthy population, limited the possibility of making adjusted comparisons between groups.
However, we recognize that this is a priority for future phases of the study, so we are currently working on expanding the cohort with the goal of achieving an adequate n per subgroup, allowing for meaningful multivariate analyses and exploring the predictive potential of GLX or GLN/GLU on clinical outcomes, as you suggest.
Regarding the interpretation of the observed patterns (glutamate peak on day 7, progressive increase in glutamine, and the behavior of GLX and GLN/GLU), we clarify that the aim is not to establish a direct causal relationship at this stage, but rather to propose a pathophysiological hypothesis based on previous literature, primarily studies in the adult population with TBI of varying severity.
Sustained elevation of plasma glutamate has been associated with greater subsequent damage after the TBI event in studies in the adult population with moderate and severe TBI, where such damage is considered to be associated with excitotoxicity, mitochondrial dysfunction, and alterations in cerebral autoregulation (Bullock et al., 1998; Timofeev et al., 2011). Although these phenomena have been rarely described in the pediatric population with mild TBI, the observed trend suggests a possible activation of similar mechanisms, even in the absence of severe structural damage. In this sense, rather than speculative, the interpretation we present seeks to open scientific discussion on the role of excitatory amino acids as early biomarkers or predictors of subclinical evolution in mild TBI, particularly in pediatrics, where neurodevelopment is a key factor.
- Back to point # 2, The study does not benchmark the proposed amino-acid measures (GLU, ASP, GLN, GLY; GLN/GLU; GLX) against established TBI protein biomarkers (e.g., GFAP and UCH-L1 for acute/astroglial–neuronal injury; S100B in pediatrics; NfL and tau for axonal/subacute injury). With out head-to-head comparisons, it is difficult to judge the incremental diagnostic or prognostic value of these metabolites or to position them within existing clinical pathways.
ANSWER:
We appreciate your observation, returning to point 2. It is true that the study does not establish direct comparisons between the proposed amino acids (GLU, ASP, GLN, GLY; GLN/GLU; GLX) and recognized protein biomarkers for traumatic brain injury (TBI), such as GFAP, UCH-L1, S100B, NfL, or tau. This omission makes it difficult to assess the incremental diagnostic or prognostic value of amino acids within the currently considered clinical pathways. However, it is important to contextualize this deficiency within the framework of the existing limitations with protein biomarkers, especially in the pediatric population with TBI.
Conventional tools such as computed tomography (CT), magnetic resonance imaging (MRI), and neurological assessments have limited sensitivity for detecting alterations in mild TBI, particularly in children. Despite the recognition of proteins such as GFAP, UCH-L1, NSL, and especially SB100 have been analyzed as biomarkers of injury in pediatric TBI, there are still no clinical guidelines supporting their routine use in pediatrics due to methodological barriers and a lack of solid evidence validating their effectiveness in this population (Marzano et al., 2021; Glushakova et al., 2016; Morello et al., 2024; Ganeshalingham et al., 2021; Papa et al., 2013; Filippidis et al., 2010).
Furthermore, biomarkers such as S100B, NSE, and tau have been mostly studied in adults with moderate or severe TBI, and their translation to the pediatric setting remains a challenge. Therefore, studying the role of amino acids offers an alternative and innovative approach. Glutamate (GLU), for example, has been widely linked to mechanisms of excitotoxicity after trauma, and its accumulation is associated with worse clinical outcomes. Previous studies have reported elevations in glutamate, aspartate, and glutamine in patients with severe TBI, suggesting that the amino acid profile could sensitively reflect trauma-induced neurochemical alterations. Although direct comparative studies with established biomarkers are essential, the metabolomic approach may offer significant advantages: it is less invasive, allows for longitudinal monitoring, and could be especially useful in the pediatric population, where rapid detection of neurological damage is crucial. Therefore, although the lack of comparisons limits the interpretation of the clinical value of these amino acids, the current findings open a promising avenue for future research integrating both types of markers into more comprehensive diagnostic and prognostic strategies. We modified a paragraph in the Introduction section, lines 99-103 and Conclusions section lines 442-445.
1: Marzano LAS, Batista JPT, de Abreu Arruda M, de Freitas Cardoso MG, de Barros JLVM, Moreira JM, Liu PMF, Teixeira AL, Simões E Silva AC, de Miranda AS. Traumatic brain injury biomarkers in pediatric patients: a systematic review. Neurosurg Rev. 2022 Feb;45(1):167-197. doi: 10.1007/s10143-021-01588-0.
2: Glushakova OY, Glushakov AV, Hayes RL. Finding effective biomarkers for pediatric traumatic brain injury. Brain Circ. 2016 Jul-Sep;2(3):129-132. doi: 10.4103/2394-8108.192518.
3: Morello A, Schiavetti I, Lo Bue E, Portonero I, Colonna S, Gatto A, Pavanello M, Lanotte MM, Garbossa D, Cofano F. Update on the role of S100B in traumatic brain injury in pediatric population: a meta-analysis. Childs Nerv Syst. 2024 Nov;40(11):3745-3756. doi: 10.1007/s00381-024-06565-8.
4: Ganeshalingham A, Beca J. Serum biomarkers in severe pediatric traumatic brain injury-a narrative review. Transl Pediatr. 2021 Oct;10(10):2720-2737. doi: 10.21037/tp-20-386.
5: Papa L, Ramia MM, Kelly JM, Burks SS, Pawlowicz A, Berger RP. Systematic review of clinical research on biomarkers for pediatric traumatic brain injury. J Neurotrauma. 2013 Mar 1;30(5):324-38. doi: 10.1089/neu.2012.2545.
6: Filippidis AS, Papadopoulos DC, Kapsalaki EZ, Fountas KN. Role of the S100B serum biomarker in the treatment of children suffering from mild traumatic brain injury. Neurosurg Focus. 2010 Nov;29(5):E2. doi: 10.3171/2010.8.FOCUS10185.
Round 2
Reviewer 3 Report
Comments and Suggestions for Authors
Accepted in the present form